# Plexiform Fibromyxoma in the Stomach: Immunohistochemical Profile and Comprehensive Genetic Characterization

**DOI:** 10.3390/ijms25094847

**Published:** 2024-04-29

**Authors:** Annabella Di Mauro, Rosalia Anna Rega, Maddalena Leongito, Vittorio Albino, Raffaele Palaia, Alberto Gualandi, Andrea Belli, Imma D’Arbitrio, Pasquale Moccia, Salvatore Tafuto, Annarosaria De Chiara, Alessandro Ottaiano, Gerardo Ferrara

**Affiliations:** 1Pathology Unit, Istituto Nazionale Tumori, IRCCS Fondazione “G. Pascale”, 80131 Napoli, Italy; rosalia.rega@istitutotumori.na.it (R.A.R.); a.gualandi@icloud.com (A.G.); imma.darbitrio@istitutotumori.na.it (I.D.); p.moccia@istitutotumori.na.it (P.M.); gerardo.ferrara@istitutotumori.na.it (G.F.); 2Department of Gastro-Hepato-Pancreato-Biliary Surgery, Istituto Nazionale Tumori, IRCCS Fondazione “G. Pascale”, 80131 Napoli, Italy; maddalena.leongito@istitutotumori.na.it (M.L.); v.albino@istitutotumori.na.it (V.A.); r.palaia@istitutotumori.na.it (R.P.); a.belli@istitutotumori.na.it (A.B.); 3Sarcomas and Rare Tumors Unit, Istituto Nazionale Tumori, IRCCS Fondazione “G. Pascale”, 80131 Naples, Italy; s.tafuto@istitutotumori.na.it; 4Histopathology of Lymphomas and Sarcomas SSD, Istituto Nazionale Tumori, IRCCS Fondazione “G. Pascale”, 80131 Naples, Italy; a.dechiara@istitutotumori.na.it; 5Division of Innovative Therapies for Abdominal Metastases, Istituto Nazionale Tumori, IRCCS Fondazione “G. Pascale”, 80131 Naples, Italy; a.ottaiano@istitutotumori.na.it

**Keywords:** plexiform fibromyxoma, gastric GIST, gastric tumors, immunohistochemical and molecular study

## Abstract

Plexiform fibromyxoma (PF), also referred to as plexiform angiomyxoid myofibroblast tumor, is an exceedingly rare mesenchymal neoplasm primarily affecting the stomach. Herein, we present a case of PF diagnosed in a 71-year-old male with a history of lung cancer, initially suspected to have a gastrointestinal stromal tumor (GIST) of the stomach, who subsequently underwent subtotal gastrectomy. The histopathological and molecular features of the tumor, including mutations in *ABL1*, *CCND1*, *CSF1R*, *FGFR4*, *KDR*, and MALAT1-GLI1 fusion, are elucidated and discussed in the context of diagnostic, prognostic, and therapeutic considerations.

## 1. Introduction

Gastrointestinal stromal tumors (GISTs) comprise the majority of mesenchymal tumors in the gastric region [1]. Within the category of rare mesenchymal neoplasms affecting the stomach, a recently identified entity has been characterized as plexiform fibromyxoma (PF) by Miettinen et al. [2]. In 2007, Takahashi et al. introduced the term “plexiform angiomyxoid myofibroblastic tumor” (PAMT) to describe this novel entity [3]. Some authors [4,5] argue for the distinction between PF and PAMT as two separate entities. However, Miettinen et al., in accordance with the latest WHO classification of tumors of the digestive system, propose that the term PF be used to encompass a spectrum of mesenchymal lesions exhibiting distinctive histological features. These features range from a myxoid to collagenous stroma, with or without a myofibroblastic differentiation of neoplastic cells [1,2].

PF exhibits no significant gender predilection and is most prevalent in young to middle-aged adults, ranging from 7 to 75 years old, with a mean age of 41 years [6]. As of now, the literature reports a total of 121 documented cases of PF [7]. The clinical presentation of PF can imitate other gastrointestinal mesenchymal tumors, particularly GISTs, which are the most frequently encountered mesenchymal tumors in the gastrointestinal tract [8]. PFs are predominantly observed in the stomach, with the majority originating from the antrum and pylorus, manifesting as lobulated intramural or submucosal masses. Notably, only a singular case of PF affecting the colon has been documented in existing literature [9]. The tumor’s origin and radiological characteristics may lead to a potential misdiagnosis of GIST or other gastric mesenchymal tumors. Nevertheless, the distinct morphology, along with immunohistochemical and molecular features, play a crucial role in accurately establishing the diagnosis. The histological characteristics of PF involve a plexiform proliferation of non-atypical spindle-shaped and ovoid cells immersed within a myxoid stroma. Typically, the lesion exhibits low cellularity. To date, there have been no reported instances of metastasis or malignant transformation, although documentation exists for mucosal and vascular invasion [10,11]. The pathogenesis of PF remains largely elusive, with no well-defined molecular alterations identified. Immunohistochemical studies indicate positivity for smooth muscle actin (SMA), variable immunoreactivity for caldesmon, focal positivity or negativity for desmin, and negativity for CD117, DOG 1, S-100, and cd-34, accompanied by a low Ki-67 proliferation index [12,13,14]. In the quest for differential diagnosis, some studies have delved into genetic mutations. Notably, mutations in the C-KIT and PDGFRA genes, crucial and characteristic in GISTs, are consistently absent in all reported PF cases, reinforcing the distinction between PF and GIST [2,15,16]. Recent investigations have proposed the presence of glioma-associated oncogene homologue 1 (GLI1) in some PF tumors. GLI1 and MALAT1 mutations have been detected in a subset of PF cases, with a translocation t(11;12) (q11;q13) producing functional MALAT1 and GLI1 chimeric proteins. Additionally, GLI1 gene translocation was reported in 24% of cases, and GLI1 polysomy was reported in 8% of cases among a total of 25 cases subjected to GLI1 genetic analysis [17,18]. This translocation results in the overexpression of GLI1 protein, a phenomenon also observed in various neoplasms, activating the hedgehog signaling pathway, which is crucial in gastrointestinal development [19,20,21,22].

PF generally demonstrates a favorable prognosis, marked by the absence of recurrences or metastases, with conservative treatment through tumor resection proving to be sufficient [23,24]. However, owing to its rarity, the true biological potential of PF remains unknown, and surgery remains the cornerstone of treatment.

Given the extreme rarity of these tumors, we find it valuable for the scientific community to share every description of clinical cases that can enhance our understanding of the clinical, biological, and molecular aspects of these entities. We present a case involving a 71-year-old man who initially received a clinical diagnosis of gastric GIST at our institute. Subsequent morphological, immunohistochemical, and molecular investigations led to the revised diagnosis of gastric PF.

## 2. Case Presentation

A 71-year-old Caucasian male, previously subjected to a lobectomy for lung cancer in 2000 and presenting with cardiac comorbidities such as complete right bundle branch block, ventricular extrasystoles, and mild tricuspid insufficiency, visited our clinic in April 2014 with symptoms of melena. The patient underwent diagnostic computed tomography (CT) (Figure 1) and esophago-gastro-duodenoscopy (EGDS) at a different institution, which identified a significant gastric antrum mass measuring approximately 5 cm, indicative of GIST. However, endoscopic biopsies from the lesion yielded inconclusive results. Upon admission to our hospital, the patient underwent a repeat EGDS with endoscopic ultrasound, which identified a submucosal lesion with features indicative of GIST. Blood tests, including CEA, Ca19.9, chromogranin, gastrin, and NSE, were within normal limits. A whole-body CT scan confirmed the presence of the aforementioned lesion. Subsequently, the patient underwent a subtotal gastrectomy with D2 lymphadenectomy. The postoperative course was uneventful, and the patient was discharged on the ninth postoperative day.

Macroscopically, the surgical specimen comprised a subtotal omental gastrectomy with a stomach measuring 15 cm (major curvature) × 12 cm (minor curvature) and an omentum measuring 20 cm × 15 cm. The gastric antrum exhibited an exophytic, brownish, round, soft lesion covered by eroded mucosa with a fibrous to myxoid surface, measuring 4.5 cm in diameter. Macroscopically, the omentum appeared normal. No neoplasia was observed at the resection margins or in the loco-regional lymph nodes.

### 2.1. Histopathological Features

Microscopically, the lesion predominantly comprised oval to spindle cells embedded in a myxoid and fibromyxoid matrix, arranged in a plexiform growth pattern, accompanied by a delicate network of capillary vessels. The cytoplasm displayed high eosinophilia, and the nuclei were round to oval. Mitotic activity was virtually absent (<5/50 hpf), with a very low Ki67 proliferation index (<3%), and no cellular atypia was observed. The neoplastic proliferation affected the submucosal layers, with focal extension into the muscular wall, and was covered by eroded gastric mucosa. Immunohistochemical analysis revealed a negative cell reaction for CD117/cKIT (CD117, DAKO (Carpinteria, CA, USA)), DOG1 (K9 monoclonal antibody; Leica Microsystems, Newcastle-on-Tyne, UK), CD34 (QBEND/10, DesmBiogenex (San Ramon, CA, USA)), Desmin (D33, DAKO (Carpinteria, CA, USA)), and for S100 (S100, DAKO (Carpinteria, CA, USA)). Positive cell reactions were observed for smooth muscle actin (1A4, DAKO (Carpinteria, CA, USA)) weakly positive for caldesmon (Clone h-CD, DAKO (Carpinteria, CA, USA)), and SDHB (Ab14715, Abcam, Cambridge, UK) (Figure 2).

The morphology of cellular elements, their growth pattern, and the presence of fibromyxoid stroma, along with the immunohistochemical features, indicated a diagnosis suggestive of gastric PF.

### 2.2. Molecular Profile of Gastric PF

Mutational analysis of the *cKIT* and *PDGFRA* genes was conducted to rule out potential differential diagnoses, with no detection of cKIT or PDGFRA mutations in our samples. Subsequent comprehensive molecular testing using Illumina TSO500^®^ (San Diego, CA, USA) identified five pathogenic/likely pathogenic mutations. These mutations comprised *ABL1* p.Q472H c.1416A>T, *KDR* p.Q472H c.1416A>T, *CSF1R* p.G413S c.1237G>A, *FGFR4* p.G388R c.1162G>A, and *CCND3* p.S259A c.775T>G. Sequencing did not reveal the presence of the D842V mutation in exon 18 of the *PDGFRA* gene, nor any other mutations in exon 12 of the PDGFRA and cKIT genes. The molecular diagnosis was further confirmed by RT-PCR. To elucidate the biological role of the mutations, a Gene Ontology (GO) analysis was undertaken, revealing associated GO terms for each mutation in a tabular format (Table 1).

The identified mutations implicated involvement in cell proliferation, differentiation, and angiogenic pathways. The tumor mutational burden (TMB) was low (3.5 muts/Mb). Cytogenetic analysis of FFPE tissue was utilized to detect the MALAT1-GLI1 translocation, which is indicative of a subset of PF. Although the occurrence of MALAT1-GLI1 fusion is infrequent in gastric cases, its presence was verified in this instance. This confirmation was achieved using differentially labeled FISH probes targeting the *GLI1* (der 12) and *MALAT1* (der1) genomic loci. Fusion signals originating from the GLI1/12q13 and MALAT1/11q12 genes were observed in one hundred randomly selected non-overlapping nuclei, confirming the presence of the MALAT1-GLI1 fusion (der11) (Figure 3).

## 3. Discussion

PF, a rare neoplasm recently characterized by Miettinen et al. [3], was previously referred to as plexiform angiomyxoid myofibroblastic tumor (PAMT) [1]. It primarily arises in the stomach, with only one reported case affecting the colon [6,7,8]. Given its similar development site, imaging features, and macroscopic and microscopic characteristics to other mesenchymal lesions of the stomach, the differential diagnosis can be challenging. Notably, the presence of spindle cells is a common feature among various gastric tumors broadly categorized as “spindle cell tumors of the gastrointestinal tract” [9]. However, PFs can be differentiated from other spindle cell tumors of the stomach by the absence of S-100 protein staining (unlike schwannomas), c-Kit, and DOG1 (unlike GISTs). Typically, tumor cells show positivity for vimentin, SMA, and H-caldesmon, while being negative for S-100 protein and c-KIT, supporting the diagnosis of PF [3].

Our patient initially presented with a suspected diagnosis of GIST based solely on radiographic and endoscopic findings. To precisely define the histomorphological features of the lesion, we conducted an immunohistochemical study. The panel revealed strong positivity for SMA, caldesmon, SDHB, and CD34, moderate positivity for vimentin, and negativity for cKIT, DOG1, S-100, and desmin. These findings alone could suffice to rule out a diagnosis of gastric schwannoma and GIST.

However, contrary to most literature findings [3], neoplastic cells in our case exhibited strong positivity for CD34, a marker typically expressed by a subgroup of GISTs [10], without expression of CD117/c-Kit and DOG1. It is imperative to address the molecular findings that unequivocally ruled out a GIST. In fact, unlike PF, patients with GISTs often confront a grim prognosis due to frequent recurrence and resistance to traditional chemotherapy and radiotherapy regimens [14,19]. GIST tumors typically harbor multiple activating mutations in the *cKIT* and *PDGFRA* genes, frequently localized to specific domains like the juxtamembrane domain (exon 11) and the extracellular domain (exon 9), with rare occurrences in kinase domains (exons 13 and 17). In *PDGFRA*, prevalent mutations occur in exons 12 and 18, with sporadic cases found in kinase domains (exons 13 and 17), as well as exons 12, 14, and 18. Our mutational analysis aimed to detect alterations in *cKIT* and *PDGFRA* genes in our sample, yet no mutations were detected, thus conclusively ruling out GIST.

To delineate the mutational landscape of our PF case and pinpoint genetic aberrations driving tumorigenesis, we conducted comprehensive molecular investigations using the TSO500 NGS panel and examined the presence of the MALAT1-GLI1 fusion via FISH. Significantly, DNA sequencing unveiled missense mutations in *ABL*, *CCND3*, *CSF1R*, *KDR*, and *FGFR4*, potentially contributing to tumor progression and angiogenesis in PF. Furthermore, the identification of the GLI1 fusion arising from a recurrent MALAT1-GLI1 translocation characterizes a distinct subset of PFs marked by hedgehog (HH) signaling pathway activation [17]. GLIs play a central role in HH signaling, exerting abnormal activation associated with tumorigenesis. GLI1 has been shown to upregulate the expression of genes involved in various cellular functions, including cell proliferation (cyclin) and angiogenesis (VEGF family) [25,26]. Our analysis demonstrated mutated expression of key components of the HH pathway, including GLI1 by FISH, along with transcriptional targets such as CCND3 and FGFR4 (as detected by NGS). Unfortunately, RNA degradation in this case precluded molecular confirmation of the *GLI1* fusion.

## 4. Conclusions

In conclusion, an accurate histomorphological review, supported by a specific immunohistochemical panel and molecular investigations, is the essential approach for confirming the diagnosis of PF and ruling out other mesenchymal neoplasms. This study represents, to our knowledge, the first report of a PF demonstrating specific genetic mutations revealed by NGS. It is noteworthy that the mutations detected in *ABL1*, *KDR*, *CCND1*, *CSF1R*, and *FGFR4* present promising therapeutic targets, with increasingly specific and potent inhibitors expected to become available in the near future. However, a detailed discussion of the pharmacokinetics and pharmacodynamics of these molecules exceeds the scope of this article. Further research involving larger cohorts is necessary to delve deeper into the molecular background and identify robust biomarkers for PF.

## 5. Materials and Methods

### 5.1. Molecular Analysis by RT-PCR

Mutational analysis of the cKIT and PDGFRA genes was performed to exclude differential diagnoses. The most representative FFPE sample was selected; the tumor area was localized on an H/E slide and subjected to microdissection. Genomic DNA was then isolated (QIAamp DNA FFPE Tissue Kit, Qiagen Inc., San Diego, CA, USA) and amplified (PCR) for exons 9, 11, 13, and 17 of the KIT gene and exons 12 and 18 of the PDGFRA gene. All PCR amplification products were directly sequenced by an automated fluorescence cycle sequencer (ABIPRISM 3130, Applied Biosystems, Foster City, CA, USA). No cKIT or PDGFRA mutations were detected in our samples.

### 5.2. Illumina TruSight Oncology 500 Assay

For further analysis of this rare case, the Illumina’s commercial TruSight Oncology 500 assay was used. This panel assesses both DNA and RNA. The DNA assay scans 523 genes for single nucleotide variants (SNVs) and indels, and the RNA assay scans 55 genes. The full list of genes can be found on the manufacturer’s website (https://www.illumina.com/content/dam/illumina-marketing/documents/products/gene_lists/gene_list_trusight_oncology_500.xlsx, accessed on 21 September 2023). It can also assess both microsatellite instability (MSI) and tumor mutational burden (TMB). The DNA and RNA extracted from FFPE tissue using the Allprep DNA/RNA FFPE Kit (Qiagen Inc.) were converted to cDNA. To enrich the library pool, probes with hybridized DNA and cDNA were magnetically attracted to streptavidin-coated beads and eluted. Library normalization was performed using a simple bead-based protocol, followed by pooling and sequencing on NextSeq 500 instrument (Illumina^®^, San Diego, CA, USA). DNA and RNA data were analyzed using Illumina TSO500 Local App Software v1.3.1 and TST170 Local App Software v1.0.1, respectively, and a customized analysis pipeline within the Clinical Genomics Workspace software (10-10-2023) platform provided by PierianDx (San Diego, CA, USA). A custom variant filter was set to include only nonsynonymous coding variants with read depth >50, excluding benign variants according to ClinVar database [13]. The remaining variant subset was visually inspected and suspected artefacts were excluded.

### 5.3. Interphase FISH

Interphase dual-color FISH was performed on tumor sections using the break-apart approach to investigate the prevalence of the *MALAT1-GLI1* fusion. By using differentially labelled FISH probes, which were mapped to the *GLI1* and *MALAT1* genomic loci, the presence of the *MALAT1-GLI1* fusion was confirmed in this case. The BAC probes used were the SG-labelled RP11-472D15 and the SO-labelled RP11-181 L23, covering the *MALAT1* and *GLI1* genes. FFPE 3.5 µm sections were taken from the case under investigation and subjected to interphase FISH in order to screen for rearrangements involving the genes of interest. Tissue sections were processed using the ZytoLight FISH Tissue Implementation Kit (Zytomed Systems, Berlin, Germany) in combination with differentially tagged BAC probes, SpectrumOrange (SO) and SpectrumGreen (SG), covering the *GLI*-1/12q13 and MALAT-1/11q12 locus regions, according to the manufacturer’s instructions. FISH was performed according to standard procedures. FISH images were captured using a CytoInsight GSL 10 fluorescence microscope (Leica Biosystems, Solihull, UK) with DAPI, SpectrumGreen, and SpectrumOrange optical filters and a 63× objective (oil, 1.30 aperture; Leica, Solihull, UK) equipped with the CytoVision digital image analysis system. Tissue was scored and considered positive if >25% of at least 200 cells showed evidence of cleavage or fusion. Excluded from evaluation were nuclei with an incomplete set of signals.

## Figures and Tables

**Figure 1 ijms-25-04847-f001:**
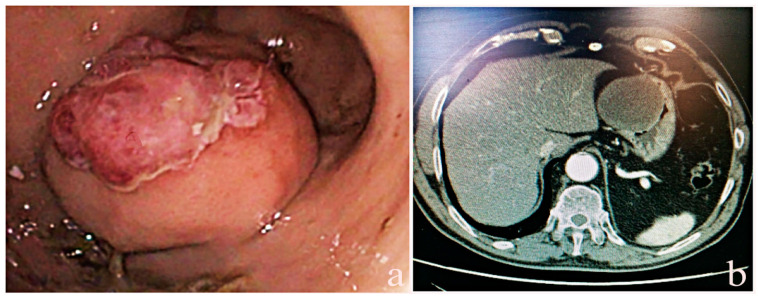
Gastric plexiform fibromyxoma case: (**a**) endoscopic appearance of antral mass with surface erosion/ulceration; (**b**) axial CT scan showing the lesion in which the body of the stomach is involved.

**Figure 2 ijms-25-04847-f002:**
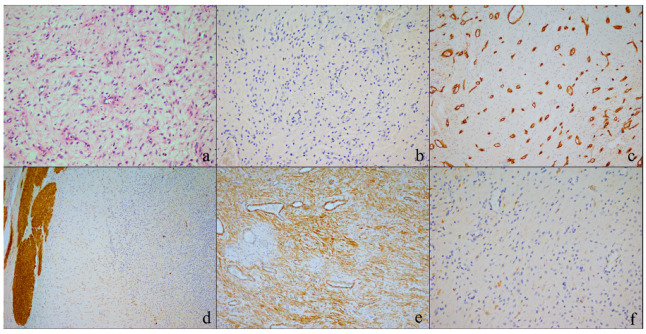
Histological findings: (**a**) plexiform fibromyxoma exhibits bland spindle cells within a myxoid matrix (hematoxylin and eosin 200×). Immunohistochemical staining showing: (**b**) negative expression of DOG1 in tumor cells (200×), (**c**) negative expression of CD34 in tumor cells, with vessels serving as an internal control (200×), (**d**) negative expression of desmin in tumor cells, with the smooth muscle layer acting as an internal control (100×), (**e**) positive expression of SMA in tumor cells (100×), (**f**) tumor cells are weakly positive for caldesmon (200×).

**Figure 3 ijms-25-04847-f003:**
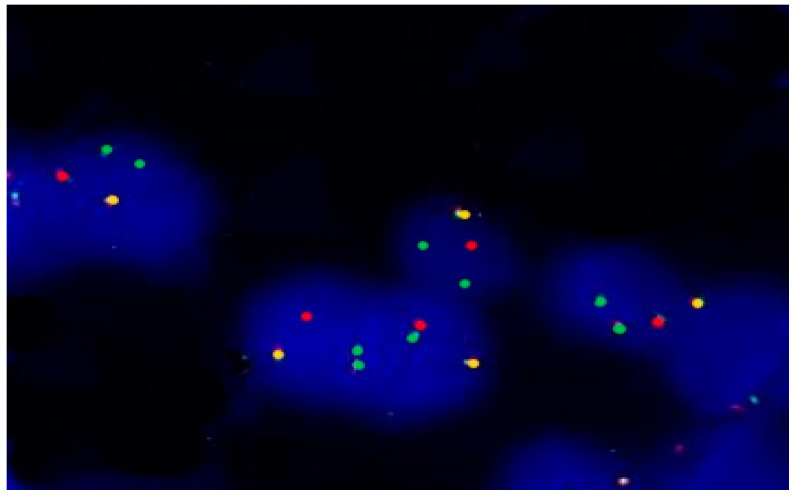
MALAT1-GLI1 fusion by FISH Interphase dual-color FISH on FFPE tissue using the SG-labelled RP11-472D15 and SO-labelled RP11-181L23 BAC probes, which cover the *MALAT1* and *GLI1* genes, respectively. Overlap of green and red signals produce yellow signals, indicating fusion of MALAT1 and GLI1.

**Table 1 ijms-25-04847-t001:** GO terms associated with mutations of PFM.

Mutations	Molecular Ontology (MF)	Cellular Component Ontology (CC)	Ontology Biological Process (BP)
*ABL1* p. c.3324A>G	Protein kinase (GO:0005524)	Cytoplasm (GO:0005737)	Phosphorylation (GO:0006468) Cell cycle (GO:0007049)
*KDR* p.Q472H c.1416A>T	Receptor-activated kinase (GO:0016301)	Plasma membrane (GO:0005576)	Angiogenesis (GO:0001525)
*CSF1R* p.G413S c.1237G>A	Protein kinase activator receptor (GO:0004713)	Plasma membrane (GO:0005887)	Signal transduction (GO:0007169)
*FGFR4* p.G388R c.1162G>A	Fibroblast growth factor (GO:0005007)	Cell membrane (GO:0005578)	Growth factor response (GO:0008283)
*CCND3* p.S259A c.775T>G	Cyclin-dependent kinase (GO:0004693)	Cell nucleus (GO:0005634)	Cell cycle (GO:0007049)

## Data Availability

Data is contained within the article.

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
