# Peer review of "Plexiform Fibromyxoma in the Stomach: Immunohistochemical Profile and Comprehensive Genetic Characterization"

_ijms, 2024, doi:10.3390/ijms25094847_

Round 1

Reviewer 1 Report

Comments and Suggestions for Authors

1. Why Plexiform Fibromyxoma was exposed? Add 5 years statistics of occurrence of the disease.

2. What is the significance of Immunohistochemical and Molecular Features in this study over other methods.

3. Language editing is required and flow of the contact should be thoroughly revised.

4. Result and discussion part need critical analysis in representing finding in better way.

Comments on the Quality of English Language

Professional editing is required 

Author Response

1. Why Plexiform Fibromyxoma was exposed? Add 5 years statistics of occurrence of the disease.

Exposing or describing a case of Plexiform Fibromyxoma (PF) is crucial due to its rarity within medical literature. By documenting individual cases, researchers can advance understanding of the disease's clinical presentation, diagnostic criteria, treatment options, and prognosis. This contributes to the accumulation of knowledge necessary for accurate diagnosis and appropriate management of future cases. The true incidence of PF remains unknown; the most accurate and transparent description of its 5 years statistics is that Plexiform Fibromyxoma is an exceedingly rare mesenchymal tumor, as previously stated in the manuscript. Thank you for these insights.

2. What is the significance of Immunohistochemical and Molecular Features in this study over other methods.

We express our gratitude to the Reviewer for their insightful comments. The significance of immunohistochemical and molecular features in this study lies in their complementary roles in cancer diagnostics. IHC is widely utilized due to its ease of implementation and reproducibility. Conversely, "molecular features", particularly comprehensive genetic profiling as indicated in the study's title, offer deeper insights into tumor characteristics by elucidating key driver genes underlying the neoplastic process. Furthermore, additional molecular characterizations are performed in the article using RT-PCR and FISH techniques. Therefore, while molecular features do not represent a specific method like IHC, they provide valuable information that enriches our understanding of the tumor's molecular landscape. In consideration of potential ambiguity in the title, a slight modification has been made to better reflect the content: "Plexiform Fibromyxoma in the Stomach: Immunohistochemical Profile and Comprehensive Genetic Characterization."

3. Language editing is required and flow of the contact should be thoroughly revised.   Language editing has been completed as suggested. Thank you.   4. Result and discussion part need critical analysis in representing finding in better way.   The points raised by the Reviewer in points 3 and 4 are interconnected. Consequently, we have undertaken language revisions, with a focus on simplifying concepts and enhancing coherence, clarity, and logical flow. These revisions primarily target the Results and Discussion sections, as the Methods section, being highly technical, is less conducive to significant modifications. We acknowledge the reviewer for their valuable insights, which have enriched the rigor and clarity of the article. While the manuscript has been revised comprehensively, more substantial changes have been highlighted in yellow.

Reviewer 2 Report

Comments and Suggestions for Authors

The authors reported a rare case of plexiform fibromyxoma in the stomach. Although this paper reports a relatively rare case and is a single case report, it is valuable in that it details the NGS results.

The only disappointment is the lack of photographs of the resected specimens. A figure of the resected specimen should be added.

Author Response

The authors reported a rare case of plexiform fibromyxoma in the stomach. Although this paper reports a relatively rare case and is a single case report, it is valuable in that it details the NGS results. The only disappointment is the lack of photographs of the resected specimens. A figure of the resected specimen should be added.

We appreciate the reviewer's recognition of our work. The patient's story begins substantially with a suspicion of GIST. The patient underwent surgery under the assumption of having a GIST. Therefore, it was not expected that the case would turn out to be so particular, so interesting for the scientific community. Consequently, no intraoperative photos were taken (which, incidentally, are not routinely performed at our institution). We deeply regret not having photos of the resected specimen. The only available in vivo image is an endoscopic one included in the mansucript. Thank you.